# Controlling Cancer Cell Death Types to Optimize Anti-Tumor Immunity

**DOI:** 10.3390/biomedicines10050974

**Published:** 2022-04-22

**Authors:** Marie Oliver Metzig, Alexander Hoffmann

**Affiliations:** 1Institute of Pathology, University Medical Center Mainz, 55131 Mainz, Germany; 2Signaling Systems Laboratory, Department of Microbiology, Immunology and Molecular Genetics, Institute for Quantitative and Computational Biosciences, UCLA, Los Angeles, CA 90095, USA; ahoffmann@g.ucla.edu

**Keywords:** immunogenic cell death, anti-tumor immunity, immunotherapy, NFκB dynamics, fate decisions

## Abstract

Over several decades, cell biology research has characterized distinct forms of regulated cell death, identified master regulators such as nuclear factor kappa B (NFκB), and contributed to translating these findings in order to improve anti-cancer therapies. In the era of immunotherapy, however, the field warrants a new appraisal—the targeted induction of immunogenic cell death may offer personalized strategies to optimize anti-tumor immunity. Once again, the spotlight is on NFκB, which is not only a master regulator of cancer cell death, survival, and inflammation, but also of adaptive anti-tumor immune responses that are triggered by dying tumor cells.

## 1. Preface

The established goal of anti-cancer therapy is to kill as many malignant cells as quickly and safely as possible so as to avoid acquired resistance and aggressive recurrence of tumors [1]. Conventional chemotherapy and radiation therapy rely on cell-suicide programs initiated by genotoxic or metabolic stress [2]. However, tumors commonly evolve mechanisms to escape [1,2]. The molecular characterization of distinct forms of regulated cell death, such as apoptosis and necroptosis, promises to inform the development of more effective, targeted, and personalized treatment strategies [3]. Novel anti-cancer agents include death receptor agonists such as tumor necrosis factor related apoptosis inducing ligand (TRAIL) [2,4], or drugs that target specific cell death pathways, such as Smac mimetics [5,6] or inhibitors of caspase activity [3,7].

Recent research progress has led to the realization that it is not only important to consider the mere elimination of tumor cells, but also the immunological consequences of the cell death process [8]. Certain types of cell death may remain immunologically silent, while others evoke a robust inflammatory response [9]. While the former may foster immune tolerance, and therefore do not support anti-cancer therapy, the latter may be immunogenic and install an effective and lasting anti-tumor immune response [9]. Combining immunogenic cell death with immunotherapy is therefore considered an attractive novel approach in combatting resistant tumors [10].

Whether inflammatory cell death is also immunogenic may depend on concurrent tumor-cell-intrinsic signaling circuits and the resulting dynamic cell fate decisions. In fact, cytotoxic cancer therapies frequently activate tumor necrosis factor (TNF) signaling and nuclear factor kappa B (NFκB) [7,11,12,13,14], a transcription factor that mediates the expression of hundreds of genes involved in cancer cell death and survival, inflammation, and immunity [15,16]. As a master regulator of inflammation, NFκB may not only regulate TNF-mediated cell fate decisions [17], but also whether dying tumor cells evoke immunogenicity [9,18,19]. Here, we review molecular circuits that determine cell death and survival decisions, and discuss their immunological consequences and potential implications for anti-tumor immunotherapy.

## 2. Different Qualities of Cell Death and Their Relevance to Anti-Tumor Immunity

The term inflammatory cell death summarizes situations in which dying cells evoke an inflammatory response [9]. Dying cells release “find me” signals in the form of damage-associated molecular patterns (DAMPs), which activate tissue-resident leukocytes and lead to the secretion of pro-inflammatory cytokines to attract more immune cells. “Eat me” signals help phagocytes to distinguish between alive and dead cells, and engulf cellular debris. Dendritic cells (DCs) mature and migrate to regional lymph nodes to cross-present the antigen to T cells [9]. Immunogenic cell death marks the possible transition from an innate to an adaptive immune response, which includes the generation of cytotoxic T cells directed against the dying cell population [9,10]. In this scenario, dying cancer cells may generate a “hot” microenvironment characterized by a pre-existing tumor-specific T-cell response, which is a prerequisite for anti-tumor immunity and successful immunotherapy [10].

Certain qualities of cell death are inherently more inflammatory and immunogenic than others and have potential relevance to tumor immunity and anti-cancer treatments (Box 1 and Table 1) [3]. Apoptosis and necrosis are two distinct death modalities with broadly distinguishable hallmarks [3].

Apoptotic cell death is the default modality of regulated cell death with physiological functions during development and tissue homeostasis [3]. It is characterized by the activation of initiator and effector caspases, followed by characteristic morphological features—shrinkage of the cell, nuclear condensation and fragmentation, and the formation of membrane-bound bodies [3]. These apoptotic remainders are typically removed by neighboring cells without evoking substantial inflammation, particularly in the absence of pathogen infection [20]. Therefore, apoptosis has long been considered non-inflammatory, and thus immunologically silent [3,9]. Indeed, caspase-mediated proteolysis may eliminate danger signals such as HMGB1 [21,22], and the exposure of phosphatidylserine (PS) facilitates the phagocytosis of apoptotic bodies [23]. Furthermore, apoptotic cells may stimulate the secretion of IL-10 and TGFβ, which suppress inflammation and may foster immune tolerance in tumors [9,24].

Necrosis was initially understood as a solely accidental event in response to pathological triggers such as traumatic injury, ischemia, or extremes of temperature [3,25]. Necrotic cell swelling and rupture of membranes lead to spilling of the cellular contents into the extracellular microenvironment, which activates cells of the innate immune system and triggers a robust inflammatory response [3,25].

Over the past two decades, regulated forms of necrotic cell death have been identified [3]. Some of these death modalities are not only inflammatory, but are also more likely than apoptosis or accidental necrosis to mediate immunogenicity and an adaptive anti-tumor response [18,19,26,27]. In this context, necroptosis or regulated necrosis is particularly relevant [9,18,19,26,27]. Necroptotic cell death shares common molecular signaling principles with apoptosis, but dying cells feature a characteristic necrotic morphology [25,28]. The molecular effector of necroptosis is phosphorylated mixed lineage kinase domain-like protein (MLKL), which oligomerizes and translocates to the membrane to induce pore formation and cellular rupture [29,30].

Pyroptosis is another inflammatory form of cell death [3]. Typically triggered in cells of the innate immune system, it may be primarily described as a strategy to defend from invading pathogens [3,31]. Intracellular microbial signals initiate the formation of inflammasomes, which activate caspase 1, or alternatively caspases 4 and 5 [31]. Caspases 1, 4, and 5 cleave gasdermin (GSDMD) to generate the pore-forming domain GSDMD-N [32], which eliminates infected cells via pyroptotic cell death and further amplifies inflammation [31]. However, pyroptosis may also occur in other cell types including tumor cells [33], in response to chemotherapy [34], and induce anti-tumor immune responses [35,36].

While inflammatory cell death may be the prelude to immunity, the level of inflammation is not always proportional to the immunogenic effects that dying cells evoke [9]. In fact, despite the absence of robust inflammation, apoptotic cells may still initiate an adaptive immune response [9]. For instance, active caspases may not only degrade danger signals, but also generate neo-epitopes that function as antigens [24]. In chemotherapy-induced apoptosis, anthracyclines such as doxorubicin may cause the exposure of calreticulin, and the release of HMGB1 and ATP, which serve as adjuvants and contribute to the immunogenicity of apoptotic cells [37,38]. Although accidental necrosis is associated with robust inflammation, it fails to trigger an adaptive immune response [20,39,40,41]. In fact, hypoxia-related necrosis, which is common in the cores of aggressive tumors, may be associated with a poor prognosis [1,42,43,44]. While it is suggested that highly proliferative tumors are naturally more susceptible to hypoxia, subsequent ischemic necrosis itself may also foster tumor-promoting inflammation and immunotolerance [1,45,46].

The precise mechanisms of how dying cells mediate tumor immunogenicity are unknown. One compelling hypothesis is that pathways activated concurrently to the propagation and execution of the death signal are involved. In this context, nuclear factor kappa B (NFκB), which is frequently activated in necroptotic cells [7,17,47], may induce the expression of immunogenic gene transcription programs that boost the immune response evoked by dying cells [9,18,19]. In line with this, decoupling necroptosis from NFκB signaling abolished the immunogenic effects of dying cells [18,19]. Other studies, however, have found no evidence for NFκB mediating the immunogenicity of necroptotic cells [26,27].

The gold-standard to evaluate immunogenicity is to measure the in vivo tumor protective capacity of dying cells—animals are immunized with apoptotic or necrotic cells, followed by a later challenge with viable tumor cells [8]. It is of note that protocols of how to induce cell death broadly differ between laboratories, which may contribute to diverging results [41]. In an attempt to standardize these conditions, several studies have utilized ligand-free reductionist systems, e.g., Tet-On inducible expression or forced dimerization, to induce apoptosis or necroptosis [18,19,26,27]. However, these systems bypass endogenous signaling circuits, which may be crucially involved in determining the immunological outcomes of cell death.

In summary, the induction of immunogenic cell death may be an attractive strategy to initiate a tumor-specific T-cell response, and may set the stage for successful immunotherapy [10]. However, the molecular mechanisms of how dying cells evoke immunogenicity are still incompletely understood. In order to exploit immunogenic cell death for cancer therapy, we have to understand the intracellular signaling network and know how manipulate it in order to push cells towards a certain fate decision. We have to develop definite criteria to characterize and predict the immunological outcome of cell death triggers, and analyze their consequences for tumor immunity.

Box 1Major cell death modalities and their relevance to cancer therapy.Apoptosis is a tightly regulated process that eliminates damaged cells during development and healthy tissue homeostasis, without causing substantial inflammation [3]. It depends on the activation of caspases and is characterized by classical morphological features such as cellular shrinkage and membrane blebbing [3]. The induction of apoptosis is an important strategy in cancer treatment, but tumor cells commonly evolve mechanisms to escape [1].Necroptosis is a caspase-independent form of regulated necrosis [3]. It has been successfully triggered in several preclinical models of human cancer, and is therefore an attractive back-up strategy to treat apoptosis-resistant tumors [3]. Necroptosis is usually mediated by the kinases RIPK1 and RIPK3, and is executed by the phosphorylated mixed lineage kinase domain-like protein (pMLKL), which causes pore-formation and premature rupture of plasma membranes [3]. Necroptotic death of cancer cells may not only evoke inflammation, but also an adaptive anti-tumor immune response, which may synergize with cancer immunotherapy [3,25].Pyroptosis is another regulated form of inflammatory cell death that shares certain features with apoptosis and necrosis [48]. The pore-forming effector is gasdermin (GSDMD-N), which is generated by inflammatory caspases (usually caspase 1, or caspase 3, 4, 5, and 11 in some cell types) and is responsible for the necrotic morphology of dying cells [3,49]. Pyroptosis is typically observed in innate immune cells in response to pathogen infection, but may potentially be induced in tumor cells by several therapeutic agents [50].Ferroptosis is triggered by uncontrollable lipid peroxidation, depending on iron [3,51]. This cell death form presents with a morphology that is distinct from other types of cell death, characterized by changes of the mitochondrial appearance, including condensation and outer mitochondrial membrane rupture [51]. Similar to other types of inflammatory cell death, ferroptosis may have applications for anti-cancer treatments, which are currently being explored [52].

## 3. The Apoptosis Regulatory Network

A variety of triggers may initiate apoptotic cell death via so-called intrinsic or extrinsic pathways [2,3]. Conventional chemotherapy or radiation trigger intrinsic apoptosis and eliminate cancer cells by causing irreversible DNA damage or metabolic stress [1,2]. Anti-cancer treatments may also generate extrinsic apoptotic signals, e.g., by increasing the expression of death receptors or cytotoxic ligands, including TNF, TRAIL, or Fas ligand (FasL); if released by infiltrating immune cells or the cancer cells themselves, this may activate death receptor signaling in an auto- or para-crine manner [2,12,53].

Both the intrinsic and extrinsic pathways initiate the formation of signaling modules, which consist of multiple proteins, and sequentially recruit and activate upstream initiator and downstream effector caspases [3]. Eventually, the proteolytic activity accumulates to cross a threshold and induce the irreversible and characteristic hallmarks of apoptotic cell death (Figure 1) [54]. This threshold value is determined individually for each cancer cell by the relative abundances of pro-death and pro-survival regulators, which is also the molecular basis for heterogeneous cell fate decisions and fractional killing within monoclonal cellular populations [54,55,56]. As a result of the relatively high expression levels of inhibitory factors such as cellular FLICE-like protein (cFLIP) [57] or inhibitors of apoptosis proteins (IAP) [58], most cancer cell types require crosstalk between the extrinsic and intrinsic pathway to execute apoptosis (Figure 1).

Generally considered an inhibitor of apoptosis [59], NFκB is known to induce the expression of several anti-apoptotic proteins, including IAP, cFLIP, and Bcl-2 family members (Figure 1), which interfere with the activation of caspases in cancer cells [59,60]. Furthermore, NFκB mediates the expression of anti-oxidant genes, which inhibits apoptosis via the c-Jun N-terminal kinase (JNK) pathway [60]. NFκB may also destabilize tumor suppressor 53, a key mediator of the DNA damage response [60]. In certain scenarios, however, NFκB also has pro-apoptotic functions [59,60,61]. For instance, NFκB may induce the expression of cytotoxic ligands and death receptors, reduce the expression of anti-apoptotic target genes, or facilitate p53-mediated apoptosis [59,60,61].

Whether NFκB’s pro- or anti-apoptotic roles prevail in a particular situation may depend on the quality of the trigger and cell type-specific factors [59,60,62]. Furthermore, it may critically rely on the timing of NFκB activation. Several studies indicate that it is important to distinguish stimulus-induced from tonic or pre-existing NFκB activity [17,62]. For instance, in prostate carcinoma cells infected with Sindbis-virus, pre-infectious NFκB enhanced apoptosis, while post-infectious NFκB had no effect [62]. In cervical cancer and rhabdomyosarcoma cells, continuous doses and short pulses of TNF produced comparable fractions of apoptotic cells, supporting that stimulus-induced NFκB has only a minor effect in determining the apoptotic response [63]. Indeed, the expression of cFLIP and other anti-apoptotic regulators is only weakly inducible by TNF in most cell types [64,65], but is affected when tonic NFκB is reduced, e.g., by genetic deletion [66] or by overexpression of an inhibitory kB super repressor gene (IκB-SR) [67].

In malignant tumors, NFκB is frequently upregulated and likely contributes to therapy resistance [60,68]. Therefore, pharmacological inhibition has been proposed as a strategy to re-sensitize tumors to apoptotic therapies [59,60,66,67]. Indeed, blocking NFκB in experiments increased the sensitivity of several cancer cells to apoptosis [59,66,67], but these efforts have translated into limited clinical benefit [60]. Careful modulation with respect to signaling dynamics rather than generic inhibition may be required to convert NFκB into a pro-death signal.

## 4. The Necroptosis Regulatory Network

Similar to apoptosis, necroptosis is initiated by various cell intrinsic and extrinsic triggers, including signals that engage death receptors, pathogen recognition receptors, or nucleic acid sensors [3]. However, the best-characterized pathway of necroptosis is triggered by TNF [3,25].

In principle, TNF coordinates diverse cellular responses, which are mediated by the subsequent activation of several multi-protein complexes (Figure 2). Engagement of TNF receptor 1 (TNFR1) leads to the rapid recruitment of RIPK1 (complex I), which activates canonical NFκB to regulate the expression of hundreds of target genes involved in inflammation, cell death, and survival [15,16,69]. After a latency period, RIPK1 dissociates from TNFR1 and binds to RIPK3 (complex IIb or the necrosome), which may trigger necroptotic cell death [25,57,67,70,71,72,73]. However, if active caspase 8 is integrated, this may disassemble the necrosome and either lead to apoptotic cell death or cellular survival (Figure 2) [57]. Thus, necroptosis occurs particularly when the caspase activity is compromised [25], and is therefore an attractive strategy to treat apoptosis-resistant tumors, e.g., by combining conventional chemotherapy with caspase inhibitors [7].

Several checkpoints are in place that may transiently delay necroptotic cell death, or even mediate sustainable protection if outlasting the deadly stimulus [17,74]. This includes a tightly regulated sequence of phosphorylation and ubiquitination events, which controls the activities of complex I and the necrosome [74,75,76,77]. Furthermore, downstream of the active necrosome, the ESCRT-III mechanism continuously counteracts pMLKL and eliminates damaged portions of the plasma membranes [78]. Thus, the necrosome may take several hours to form, propagate, and produce sufficient amounts of pMLKL to irreversibly induce cell death [47,70,72,73,79].

Some of these checkpoint mechanism are NFκB-responsive, i.e., they are affected by known NFκB target genes (Figure 2) [47,75]. For instance, TNFAIP3/A20 restricts TNF-mediated apoptotic and necroptotic cell death by stabilizing M1-ubiquitin chains in upstream complex I [80,81,82,83]. In addition, A20 specifically inhibits necroptosis by de-ubiquitinating RIPK3, thus restraining the necrosome [17,84]. IAP proteins cIAP1 and cIAP2 mediate the K63-linked ubiquitination of complex I and thus cellular survival [58,85,86], while the cylindromatosis (CYLD) gene restricts M1-linked ubiquitination and facilitates TNF-induced cell death [75]. The stability of the necrosome is regulated by cFLIP relative to caspase 8 present in complex II [57,87].

Elevated tonic NFκB in cancer cells may not only mediate apoptosis resistance [59,60,66,67,68], but also sustainably protect from necroptosis [17]. In contrast, NFκB that is induced by drug treatments has been shown to augment necroptosis when inducing the production of autocrine TNF by cancer cells [7,11,29].

The timing of NFκB activation may not only determine whether cancer cells will survive or die, but also affects the quality of cell death induced by a given treatment. For instance, de novo A20 that is expressed in response to TNF may integrate into the necrosome, but less so into complex I, thereby specifically protecting from necroptosis, but not from apoptosis [17,84]. This protection, however, is only transient—when inducible NFκB activity subsides, but the deadly stimulus persists, cells will still undergo necroptosis, but in a delayed manner [17]. In turn, one may speculate that this delay grants cells time to execute NFκB-dependent gene expression programs. These programs may not only affect the quantity, quality, and dynamics of cell death responses, but also the immunogenicity of dying cells [9,17,18,78]. In contrast, tonic NFκB activity may sufficiently protect cancer cells from apoptosis, but instead switch the bias of the regulatory network to rather inflammatory modalities of cell death. In fact, NFκB is a well-known priming signal for the assembly of the inflammasome [49], which is upstream of caspase 1 activation. Caspase 1 not only induces the processing and release of pro-inflammatory IL-1β and IL-18, but also cleaves GSDMD, the molecular effector of pyroptosis [60,68].

Targeting NFκB-responsive regulators and their expression kinetics may therefore offer novel therapeutic strategies to optimize cytotoxic therapies, modulate between inflammatory and non-inflammatory cell death modalities, and exploit the immunogenic effects of dying cells.

## 5. Determinants of Whether Inflammatory Cell Death Is Immunogenic, and NFκB’s Role in It

The minimal requirement for an anti-tumor immune response is an immunogenic antigen in the presence of adjuvants, both of which dying cells are able to provide. A specific antigen may be expressed in cells before cell death is triggered, and the mode of antigen delivery, e.g., cellular necrosis, may then drive an inflammatory response. In this scenario, the mere extent of tissue damage would be the main determinant of whether immunogenicity is established or not. However, even in the presence of appropriate antigen and robust inflammation, immunogenicity is only a potential rather than an inevitable consequence of cell death. Then, what are the additional determinants of whether cell death results in an immune response?

While the individual’s specific immune status clearly matters, immunogenicity likely requires a precise composition of the released antigens and adjuvants, which are affected by the initiation and execution of cell death itself. Rather than being passive corpses, dying cells actively generate signals that are directed towards antigen-presenting cells, and impact innate and adaptive immune responses [9,24]. “Find me” and “eat me” signals regulate how efficiently immune cells are attracted and at what rate the antigens are taken up, processed, and displayed. Phagosomes may not only contain antigens, but also other molecules that influence how DCs process the antigen and maturate. These signals will shape subsequent T-cell responses and ultimately decide whether immunogenicity is established.

Active caspases during apoptosis have been shown to deactivate DAMPs and destroy antigens [21,22], but may also generate neo-epitopes that are specifically recognized by T cells during cross-priming [24]. Active necroptosis signaling involves the formation of amyloid-like protein complexes, e.g., RIPK3-RIPK3 oligomers [70], which may evoke innate and adaptive immune responses [9,88]. These complexes may be actively delivered prior to cellular disintegration, e.g., via the ESCRT mechanism and the exosome pathway [78], or after necrotic bursting via the phagocytosis of cellular debris [9].

A compelling hypothesis is that immunogenicity also depends on signaling that is activated alongside the primary death-inducing pathway within dying cells [9]. In necroptotic cells, for instance, concurrent NFκB may be required for the mediation of immunogenic gene expression programs [18,19], including the coordinated release of pro-inflammatory cytokines [18,19,89]. While the exact mechanisms remain to be elucidated, NFκB may also induce the expression of neoantigens or genes that influence the innate and adaptive immune response. Similar mechanisms may extend to immunogenic forms of apoptosis, especially as the reported triggers such as poly I:C [18,90] or doxorubicin [37,38,91] are known to also activate NFκB [13].

Thus, cytotoxic agents that trigger the NFκB-dependent gene expression are promising substances to induce immunogenic cell death in cancer. These include Smac mimetics, which directly target the inhibitors of apoptosis proteins (IAP) [5], therefore affecting the canonical and non-canonical NFκB activity [58,92], and possibly the immunogenicity of cell death. The chemotherapeutic 5-fluorouracil, in combination with pan-caspase inhibitors, induces necroptosis and activates NFκB, likely via genotoxic stress-related mechanisms [7]. Several anti-cancer drugs also induce the secretion of TNF [7,11,12,14]; auto- and paracrine TNFR signaling may not only synergize with apoptotic or necroptotic death, but also activate immunogenic NFκB signaling in tumor cells [7,17,67].

Active NFκB in dying cells may also be the result of signaling crosstalk with components of certain cell death pathways. For instance, pro-caspase 8 [93] or RIPK3 [94,95] are capable of activating NFκB, which may be particularly relevant for necroptotic drug treatments [89]. Interestingly, it has been postulated that alternative functions of RIPK3 [96], such as activation of the NFκB-responsive gene expression, may even be sufficient to confer anti-tumor immunity, independently from whether necroptotic cell death is executed or not [19]. Similarly, the release of cytokines such as IL1β downstream of inflammatory caspases may confer anti-tumor immune responses, regardless of whether the majority of tumor cells initially undergo pyroptosis. In fact, one recent study demonstrated that pyroptosis in less than 15% of cells within engrafted tumors was sufficient to increase the number of tumor infiltrating lymphocytes and induce effective anti-tumor immunity [36].

## 6. Dynamics of Cell Fate Decisions as a Determinant of Immunogenic Cell Death

Molecular networks, as described in Figure 1 and Figure 2, lay the basis for a mechanistic understanding of cancer cell death and survival decisions. Dynamic activities of pro-survival and pro-death regulators such as NFκB critically determine the quality and rate of cell death decisions (Figure 3). Whether cancer cells die rapidly after receiving the stimulus, or take a latency period before cellular disintegration may alter gene expression profiles, the composition of the secretome, and ultimately have an impact on the immunogenic consequences of dying cells (Figure 3).

It is of note, however, that cell fate decisions are heterogeneous. Not all cells die at the same time, and some cells may even survive in response to a cytotoxic stimulus; this phenomenon is known as fractional killing [54,55,97,98,99]. The origins of heterogeneous cell fate decisions can be genetic or non-genetic. In tumors, chromosomal instability or acquired genetic mutations under selection pressure can lead to the emergence of subclones that may be more resistant to cytotoxic stimuli [1]. However, even in monoclonal cancer cell populations, stochastic differences in gene expression and other processes can lead to varying abundances of pro-survival and pro-death regulators [55]. These differences can pre-exist and could be determined by a cell’s history, or be inducible, for instance, by the cytotoxic stimulus itself [17,55]. Cell-to-cell variability may affect the overall quantities of cell death, i.e., fractional killing in response to an anti-cancer drug, and thus the quantities of immune stimulatory molecules that are released by a cellular population. It may also lead to mixed qualities of cell death that occur in response to the same stimulus within a population: some cells may undergo apoptosis, while others may die of necrotic forms of cell death. In addition, cell-to-cell heterogeneity may influence the dynamics of cell death decisions within a population [17]. When a necroptosis-susceptible population of fibrosarcoma cells is treated with TNF, for instance, this will induce rapid death within one subpopulation, while another subpopulation initially survives. While transient survival is the result of TNF-induced activation of NFκB and the subsequent expression of pro-survival A20 [17], the vast majority of cells will eventually die if the TNF stimulus persists and NFκB activity subsides [17]. The resulting two-phased death kinetics within cellular populations may have immunological consequences—while rapidly dying cells may predominantly deliver pre-existent DAMPs and antigen, transiently surviving cells will be granted time to execute concurrent NFκB-dependent gene expression programs. In tumors, these death kinetics and both the surviving and dying cellular fractions may orchestrate the innate and adaptive immune response, and ultimately decide whether the host responds with inflammation, immunogenicity, or immunotolerance (Figure 3).

## 7. Outlook: The Targeted Induction of Immunogenic Cell Death in Cancer Therapy

Despite rapid developments in the field of personalized medicine, therapy resistance, relapse, and poor survival remain common challenges in the treatment of cancer patients [1]. While conventional chemotherapies have rather generic mechanisms of action and commonly produce damage in healthy tissues with potentially severe side effects, targeted therapeutic substances such as death receptor agonists promise to induce apoptosis more selectively in malignant cells [2]. Furthermore, the discovery of alternative forms of regulated cell death, including necroptosis, pyroptosis, and ferroptosis, has opened up exciting avenues to overcome apoptosis-resistance in cancer [3].

Immunotherapy such as immune checkpoint inhibition has been proven to be successful in malignant melanoma and other solid tumors, but so far it is not available for the majority of cancer patients [10]. Resistance has been attributed to an insufficient mutational landscape, and thus a lower probability of cancer cells to be recognized by T cells, resulting in immunologically “cold” tumors [10]. However, in several tumor entities, T-cell infiltration is associated with better prognosis despite a low mutational burden [68,100,101]. This supports a level of optimism that it is possible to convert “cold” into “hot” tumors, and to treat these patients with immune checkpoint inhibitors [10,68].

The targeted induction of immunogenic cell death may present a strategy to render “cold” tumors accessible to immunotherapy [9,10]. Antigen and immune stimulatory molecules delivered by dying cells may trigger pre-existing tumor-specific T-cell responses, which is a prerequisite for immune checkpoint inhibition [9,10]. Inflammatory modes of cell death such as necroptosis and their triggers, e.g., Smac mimetics or 5-FU in combination with pan-caspase inhibitors, may be particularly effective in combination with immune checkpoint inhibition.

However, inflammation caused by cytotoxic therapies is not a guarantor of an effective anti-tumor immune response [9]. It may even have opposite, namely tumor-promoting, effects [102].

Whether inflammation is beneficial or detrimental may be determined to some extent by the trigger itself, as well as the quality and quantity of the cell death response. However, it may also depend on tumor cell-intrinsic signaling circuits, such as NFκB, and the subsequent dynamics of cell fate decisions.

NFκB is a well-established master regulator of apoptosis in tumor cells, and overwhelming inflammation associated with aberrant NFκB is linked to carcinogenesis, tumor progression, and therapy resistance [102]. Thus, generic inhibition of IKK/NFκB has been explored as a treatment strategy, but has shown limited clinical benefit [59,60,102]. More recently, NFκB was shown to regulate inflammatory cell death and possibly anti-tumor immunity [17,18,19]. Whether therapy-induced NFκB may primarily confer tumor immunity or hamper the efficient execution of cell death is likely context dependent.

A systematic understanding of NFκB-responsive cell fate decisions is of utmost clinical interest in order to exploit immunogenic cell death as a treatment option in resistant cancer. Deciphering the molecular network that encodes dynamic cell death and survival decisions bears the potential to predict whether tumor cells are primed for apoptosis or necroptotic cell death, and to identify potential biomarkers and therapeutic targets to push tumor cells towards one or the other. A recent discovery that human cancer cells more reliant on mitochondrial respiration are particularly sensitive to copper-induced cytotoxicity demonstrates the need for metabolic tumor profiling [103]. Tissue-based biomarker studies are urgently needed to link expression signatures to an inherent predisposition for immunogenic cell death. Thus, we call for clinical cancer therapeutic studies that include pre- and post-treatment immunophenotyping [104], and relate immune activation to NFκB activities and the types of cell death present in tumor tissues.

## Figures and Tables

**Figure 1 biomedicines-10-00974-f001:**
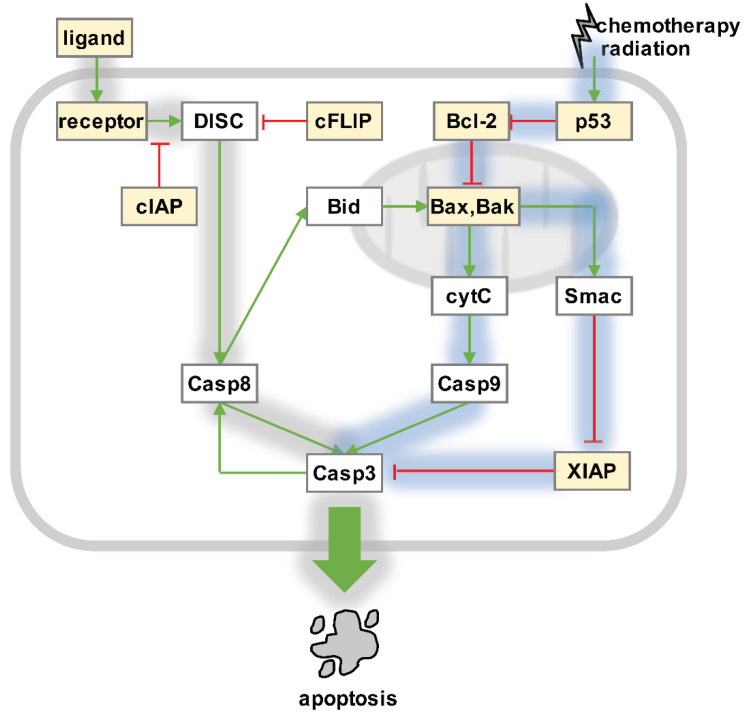
Apoptosis signaling. Intrinsic pathway (blue shading): DNA damage is sensed by tumor suppressor p53. P53 controls the Bcl-2 protein family, which includes pro-apoptotic (e.g., Bid and Bax) and pro-survival (e.g., Bcl-2) factors that tightly regulate mitochondrial outer membrane permeabilization (MOMP). Activated Bax and Bak form pores in the outer mitochondrial membranes, which allow Smac/Diablo and cytochrome c to translocate from the intermembrane space into the cytosol. Cytochrome c binds Apaf-1 and caspase 9, which form the so-called “apoptosome” to activate effector caspases 3, 6, and 7. Smac/Diablo inhibits XIAP, which releases the block on the proteolytic activity of effector caspases. Extrinsic pathway (gray shading): Pro-apoptotic ligands such as TNF, FasL, or TRAIL engage death receptors on tumor cells, leading to the formation of death-inducing signaling complex (DISC) by recruiting FADD and the pro-forms of initiator caspases 8 and 10. Active initiator caspases may then activate effector caspases 3, 6, and 7, but this is limited in most cancer cells because of inhibitory factors such as cIAP, cFLIP, and XIAP. Caspase 8 may engage the intrinsic pathway via Bid truncation and MOMP. Released effector caspases may create a positive feedback and activate more caspase 8 in some cells. NFκB-responsive regulators (yellow boxes): The molecular network of apoptosis regulation includes NFκB target genes on several levels. Changes in NFκB activities therefore determine the abundances of pro-survival and pro-apoptosis regulators, and affect tumor cell fate decisions in response to death-ligands, chemotherapy, and radiation.

**Figure 2 biomedicines-10-00974-f002:**
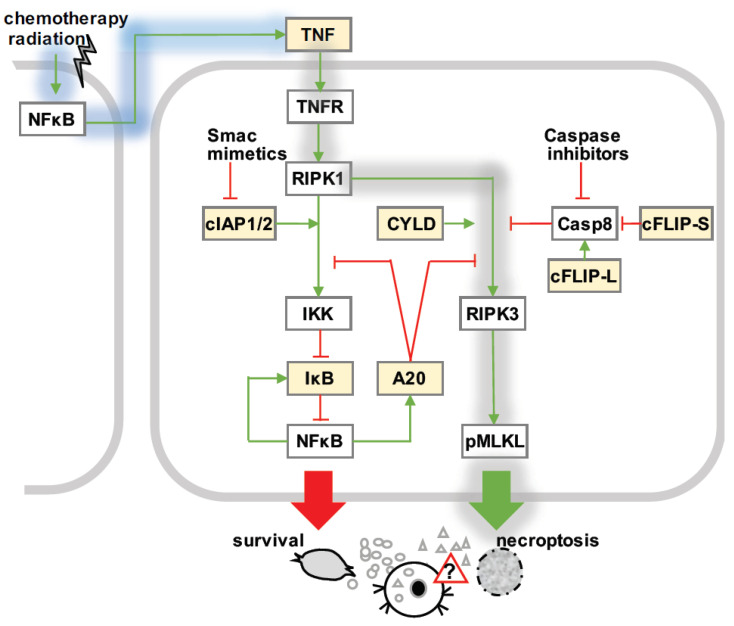
Necroptosis signaling. TNF-mediated necroptosis signaling (gray shading): Upon TNF stimulation, rapid recruitment of RIPK1 to TNFR1 (complex I) leads to canonical activation of NFκB via the inhibitor kB kinase (IKK), which targets the inhibitor of κB (IκB) proteins for phosphorylation and subsequent degradation. After a delay period, RIPK1 dissociates from plasma-membrane-bound complex I to bind to RIPK3 (complex IIb or necrosome). Activated RIPK3 recruits and phosphorylates MLKL (pMLKL), which executes necroptotic cell death. cIAP1 and cIAP2 stabilize complex I, while CYLD facilitates the formation of the necrosome. A20 may hamper the activation of IKK via complex I, or inhibit the activation of RIPK3 in the necrosome. The ratio of long and short cFLIP isoforms relative to caspase 8 control the activity of caspase 8: pro-caspase-8-cFLIP-L, but not –cFLIP-S heterodimers destabilize the necrosome. Necroptosis is facilitated by caspase inhibitors, and Smac mimetics, which target cIAP1 and cIAP2 for degradation. NFκB-responsive regulators (yellow boxes): Basal and inducible NFκB is not only known to control the gene expression programs regulating necroptosis decisions (yellow), but may also affect the immunogenicity of dying cells. Cancer therapies trigger NFκB via DNA damage and initiate auto- or para-crine TNF signaling, which mediates tumor cell survival or death.

**Figure 3 biomedicines-10-00974-f003:**
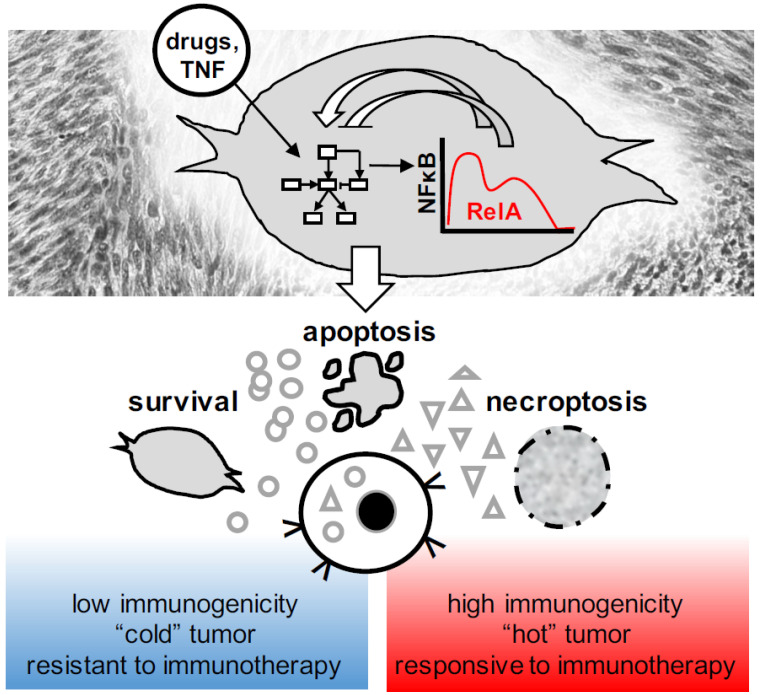
Dynamic apoptosis and necroptosis decisions and their consequences for tumor immunity. NFκB is a dynamic regulator of the molecular network that governs over cell death and survival, as well as the immunological consequences of dying tumor cells. Immunogenic cell death induced by anti-cancer therapies may convert a subset of “cold” tumors into “hot” tumors, and thus help to overcome resistance to immunotherapy.

**Table 1 biomedicines-10-00974-t001:** Different cell death modalities with characteristic features.

	Key Inducers	Initiation and Execution	Rupture of Membranes	Immunological Consequences
Apoptosis	death ligands (e.g., TRAIL, FasL, TNF), chemotherapy, irradiation	p53, Bcl-2 protein family, caspases 2, 3, 6, 7, 8, 9, 10	no	mostly non-inflammatory and/or immune-suppressive; immunogenic in certain situations (e.g., doxorubicin)
Accidential Necrosis	trauma, ischemia, extreme temperatures	non-regulated	yes	inflammatory
Necroptosis	TNF, chemotherapy, irradiation (preferably under caspase-deficient conditions)	RIPK1, RIPK3, MLKL	yes	inflammatory; immunogenic
Pyroptosis	microbial pathogens, chemotherapy	inflammatory caspases 1, 4, 5, caspase 3, gasdermin protein family	yes	inflammatory; immunogenic
Ferroptosis	inhibition of cell membrane transporters (e.g., system xc^−^) and antioxidant enzymes (e.g., GPX4)	iron-dependent lipid peroxidation and ROS accumulation	yes	inflammatory; possibly immunogenic

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
