# Peer review of "Controlling Cancer Cell Death Types to Optimize Anti-Tumor Immunity"

_biomedicines, 2022, doi:10.3390/biomedicines10050974_

Round 1

Reviewer 1 Report

In this review, the authors summarized the recent research progression about the relevance of cell death types and antitumor immunity. Firstly, the authors discussed apoptosis, necroptosis, pyroptosis and ferroptosis and their relevance to cancer therapies. Secondly, the role of NFκB in immunogenic cell death was mentioned. Finally, the authors claimed dynamics of cell fate decisions as a determinant of immunogenic cell death.  Overall, this review paper is interesting and meaningful. I only have some minor concerns listed below, which could be easily dealt with.

  • It will be better if the authors can use some professional software to re-make your figures. I noticed some words were underlined in red and then screen shot.
  • The authors mentioned apoptosis, necroptosis, pyroptosis and ferroptosis, but only detailly discussed apoptosis and necroptosis regulatory networks. I suggest pyroptosis and ferroptosis regulatory networks should be discussed as well.
  • A recent investigation reported a novel cell death type call “cuprotosis: which is a cell death caused by copper and targets lipoylated TCA cycle proteins. As a review study, I think this new cell death type should be mentioned.
  • It’s better to talked more about your own related work in this manuscript.

Author Response

We appreciate Reviewer #1's feedback and are happy to address all raised issues with this revised submission:

  1. It seems there may have been a compatibility problem with the figure file format. We upload PDF files this time and hope this will fix all formatting issues.
  2. We now provide a more detailed discussion on pyroptosis and ferroptosis in the text and a new Table.
  3. We thank Reviewer #1 for this valuable suggestion and are including this newly characterized mechanism of cell death in our review paper.

Reviewer 2 Report

Thank you for being selected to revise the review paper entitled: “Controlling cancer cell death types to optimize anti-tumor immunity”.

The manuscript is written in an understandable and thoughtful way. It has been divided into several major sections that describe the different characteristics of cell death, as well as, for the most part, their regulatory networks. Moreover, the authors focused on their importance in cancer therapy.

Although the discussed subject matter is not an innovative issue, the authors summarised the subject in an interesting way, citing many current studies.

The following comments are of a cosmetic nature and would increase the value of the work:

  1. Maybe it is worth compiling the information in the box regarding apoptosis, necroptosis, pyroptosis and ferroptosis in the form of a table? If the authors have an idea to summarize other information in the table - it is worth considering.
  2. I feel unsatisfied in the context of pyroptosis and its regulatory networks - I think it would be worth devoting a separate paragraph to it, considering that this is a relatively new issue and many reports openly speak about its importance in disease progression by, inter alia, inducing anti-cancer immunity.

The authors cited and described a lot of available scientific studies. The presented manuscript is the valuable and interesting summary and I recommend this paper for publication in Biomedicines journal after minor revisions.

Author Response

We thank Reviewer #2 for the positive feedback and valuable suggestions. We were happy to address all raised points in this updated version of our review paper:

  1. We added a Table to give a better overview on major modalities of cell death and potential immunological outcomes.
  2. We included a paragraph on pyroptosis in section 2 to describe the underlying regulatory network in more detail, and added information in sections 4 and 5 regarding crosstalk with NFkB and potential immunogenic consequences. Critical literature was cited to support the points.